# *Aedes* Mosquito Virome in Southwestern Cameroon: Lack of Core Virome, But a Very Rich and Diverse Virome in *Ae. africanus* Compared to Other *Aedes* Species

**DOI:** 10.3390/v16071172

**Published:** 2024-07-21

**Authors:** Karelle Celes Mbigha Donfack, Lander De Coninck, Stephen Mbigha Ghogomu, Jelle Matthijnssens

**Affiliations:** 1Laboratory of Viral Metagenomics, Laboratory of Clinical and Epidemiological Virology, Department of Microbiology, Immunology and Transplantation, Rega Institute, KU Leuven, 3000 Leuven, Belgium; 2Molecular and Cell Biology Laboratory, Biotechnology Unit, Department of Biochemistry and Molecular Biology, University of Buea, Buea P.O. Box 63, Cameroon

**Keywords:** *Aedes*, *Ae. africanus*, *Ae. albopictus*, metagenomics, core virome, eukaryotic virome

## Abstract

In Cameroon, *Aedes* mosquitoes transmit various arboviruses, posing significant health risks. We aimed to characterize the *Aedes* virome in southwestern Cameroon and identify potential core viruses which might be associated with vector competence. A total of 398 *Aedes* mosquitoes were collected from four locations (Bafoussam, Buea, Edea, and Yaounde). *Aedes albopictus* dominated all sites except for Bafoussam, where *Aedes africanus* prevailed. Metagenomic analyses of the mosquitoes grouped per species into 54 pools revealed notable differences in the eukaryotic viromes between *Ae. africanus* and *Ae. albopictus*, with the former exhibiting greater richness and diversity. Thirty-seven eukaryotic virus species from 16 families were identified, including six novel viruses with near complete genome sequences. Seven viruses were further quantified in individual mosquitoes via qRT-PCR. Although none of them could be identified as core viruses, Guangzhou sobemo-like virus and Bafoussam mosquito solemovirus, were highly prevalent regionally in *Ae. albopictus* and *Ae. africanus*, respectively. This study highlights the diverse eukaryotic virome of *Aedes* species in southwestern Cameroon. Despite their shared genus, *Aedes* species exhibit limited viral sharing, with varying viral abundance and prevalence across locations. *Ae. africanus*, an understudied vector, harbors a rich and diverse virome, suggesting potential implications for arbovirus vector competence.

## 1. Introduction

Mosquitoes (Family: Culicidae) pose a significant threat to global health as they are efficient vectors of major infectious agents [1,2]. The mosquito genera of medical importance are *Anopheles*, *Culex*, and *Aedes*, which are the most efficient pathogen vectors of the class of Arthropods [3,4]. These Genera carry pathogens (parasites, filarial worms, and arboviruses) which are responsible for at least 17% of all human and animal diseases [5,6]. Approximately 73% of these pathogenic agents are arboviruses, a significant part of which is known to originate from wildlife [7]. In Africa, arboviruses have caused over 35 arboviral diseases, and at least 26 of these diseases have been detected in Cameroon [6,8]. The most prevalent arboviral diseases in Cameroon include Dengue, Chikungunya, Yellow fever, Zika, and Rift valley fever [9,10].

In addition to arboviruses capable of infecting both vertebrate and invertebrate cells, the mosquito virome (viral part of the microbiota) also contains a large proportion of viruses which can only infect vertebrate cells and are referred to as Insect-Specific Viruses (ISVs) [11]. The main mechanism of transmission and maintenance of ISVs is through vertical transmission (from an infected female mosquito to their offspring) [12], although recent research has suggested another potential mechanism for virus transmission, which is through mosquito excreta [12,13,14]. ISVs being transferred from generation to generation in a particular mosquito population for a period of time are referred to as the “core virome”; however, a clear quantitative definition is currently lacking [11,12]. Although the exact mechanism of microbial interactions is not fully understood, ISVs such as Phasi Charoen-like virus (PCLV), Palm Creek virus and bacteria such as *Wolbachia* have been hypothesized to modulate their mosquito host with respect to vector competence for important arboviruses such as: West Nile virus (WNV), Dengue Virus (DENV), and Zika virus (ZIKV) [15,16,17,18,19].

The increased discovery of ISVs can be largely attributed to metagenomic Next-Generation Sequencing (mNGS). This valuable tool has revolutionized the identification of viruses, mosquito species, and endosymbionts, like *Wolbachia*, from minimal sample quantities [20,21]. One of the significant advantages of mNGS is its ability to detect known and unknown viruses, as most viruses are very challenging to isolate and grow. This capability has benefited mosquito virology, by unveiling a previously unexplored diversity of viruses within mosquito populations. The insights gained from viral profiling via mNGS hold significant implications for disease surveillance and public health.

In Cameroon, the most common vector of arboviral diseases is the *Aedes* mosquito [22]. Previous studies have portrayed *Ae. aegypti* and *Ae. albopictus* as the major vectors of arboviruses, with *Ae. aegypti* dominating the northern part and *Ae. albopictus* dominating the southern part of the country [6,23,24,25]. Recent studies have also shown the circulation of *Ae. africanus*, which is also very prevalent in regions where arboviral diseases have been reported; unfortunately, there is very little known about its vector competence [9,26,27]. *Ae. simpsoni* was also recently identified circulating abundantly in rural forest settings in southwestern Cameroon and in Maroua, located in North Cameroon [6,28]. The distribution of these mosquitoes varies with the prevailing climatic zones. There are five sub climatic or ecological zones in Cameroon, including Equatorial Mountain monsoon, Equatorial Guinean, Equatorial monsoon, Tropical Sudanian and Tropical Sudano–sahelian (Figure 1). The climatic zones in Cameroon are dominated by the warm desert and semi-arid climate in the north, the tropical savanna in the Central part, and the Equatorial monsoon climate in the southern part of the country and along the coast [22,23,24,28].

This study aims to elucidate the eukaryotic virome composition of the *Aedes* species prevalent in the southwestern region of Cameroon, specifically the environments of Buea, Edea, Yaoundé, and Bafoussam with a focus on identifying core viruses. To achieve this aim, we employed the mNGS technique to characterize the *Aedes* mosquito virome across these regions. Subsequently, we conducted quantitative analysis using Reverse Transcription Polymerase Chain Reaction (qRT-PCR) to investigate suspected core viruses in individual mosquitoes across different mosquito species and geographical locations. The findings of this study add valuable information on the *Aedes* eukaryotic virome composition (identity, diversity, and abundance) in southwestern Cameroon. Understanding the composition of the mosquito virome is essential as it provides fundamental knowledge to comprehending microbial interactions, which is an appealing strategy for arboviral disease control.

## 2. Materials and Methods

### 2.1. Mosquito Sampling and Processing

Mosquitoes were collected from August to September 2020 (rainy season) from farms and gardens around households at dawn and dusk in four regions in Cameroon: Buea, Edea, Yaounde, and Bafoussam (Appendix A). *Aedes* mosquitoes were trapped using the following two methods: the BG sentinel trap (supplemented by a BG lure and carbon dioxide made from yeast, water, and sugar) and aspiration using portable aspirators (Watkins & Doncaster, Cranbrook, UK). The mosquitoes were transported in a mini freezer containing ice packs to the Molecular and Cell Biology Laboratory (MCBL) of the University of Buea for characterization.

Species identification and gender determination were carried out based on morphological characteristics using the stereo binocular microscope (Leica EZ4, Wetzlar, Germany) following Pictoria Keys [29]. A total of 398 mosquitoes were captured in different climatic zones, out of which 216 female mosquitoes of dominant species were grouped into 54 pools (4 mosquitoes per pool) based on species and capture location (Table 1).

### 2.2. Viral Enrichment, Amplification, Library Construction and Sequencing

The 54 pools were processed for NGS using the NetoVIR protocol [30]. Each mosquito pool was homogenized in 400 μL PBS using 2.8 mm ceramic beads (Bertin technologies, Montigny-le-Bretonneux, France) and centrifuged at 17,000*g* × for 3 min. The supernatant (150 μL) was filtered using a 0.8 μm (PES) filter (Sartorius, Göttingen, Germany) at 17,000*g* × for 1 min. to remove bacterial and host cells and to enrich Virus-Like Particles (VLPs). Free-floating nucleic acids in the filtrate were subjected to nuclease digestion using a combination of enzymes. Specifically, 2 μL of Benzonase (25–19 units/μL) and 1 μL of Micrococcal nuclease (2,000,000 gel units/mL) were added to the sample, along with a homemade buffer (1 M Tris, 100 mM CaCl_2_, and 30 mM MgCl_2_). The mixture was incubated at 37 °C for 2 h. Subsequently, RNA and DNA were extracted using the QIAGEN Viral RNA mini kit (QIAGEN, Hilden, Germany), without carrier RNA, following the manufacturer’s instructions. Random amplification of reverse transcribed RNA and DNA was performed for 17 cycles using the Complete Whole Transcriptome Amplification Kit (WTA 2) (Sigma Aldrich, Burlington, MA, USA). The WTA2 products were then purified using the MSB^®^ Spin PCRapace (Stratec, Berlin, Germany). Libraries for Illumina sequencing were prepared using the Nextera XT DNA Library preparation kit from Illumina (San Diego, CA, USA) and further purified using 1:1 ratio of Agencourt AMPure XP beads (Beckman Coulter, Brea, CA, USA). Prior to sequencing, DNA library size distribution and quality were determined using the Bioanalyzer 2100 (Agilent Technologies, Santa Clara, CA, USA) and qubit measurements. Sequencing of the samples was performed on a NextSeq500 High throughput platform (Illumina, San Diego, CA, USA) for 300 cycles (2 × 150 bp paired ends). The target for each pool was an average of 10 million paired end reads.

### 2.3. Identification and Annotation Eukaryotic Sequences Retrieved from Sequenced Aedes Mosquito Pools

The sequencing of 54 pools generated an average of 7.3 million raw reads (0.3 to 13.6 million) per pool. The bioinformatic analysis was first conducted using the Virome Paired-End Reads (ViPER v1.1) pipeline (https://github.com/Matthijnssenslab/ViPER) accessed on 22 December 2021. First, adapters and low-quality reads were trimmed from the raw paired end reads using Trimmomatic [31]. Subsequently, reads mapping to contigs which were known to be found in reagents (contaminants) and the host genome (Accession numbers for *Aedes aegypti*: GCA_002204515.1 and *Aedes albopictus*: GCA_006496715.1) were removed using Bowtie 2 [32]. The trimmed reads were de novo assembled into 124,398 virus contigs of at least 500 bp in length using metaSPAdes v3.15.3 and clustered across samples into 57,516 non-redundant (nr) contigs using Blast v2.11.0 and CheckV v0.1.0 [33,34]. These nr contigs were taxonomically annotated using DIAMOND (v 2.0.11) (based on BLASTx), KronaTools v 2.8.1, and TaxonKit v0.9.0, which classified based on the lowest common ancestor [35,36,37]. From these nr contigs, only contigs belonging to eukaryotic viruses were extracted. Thereafter, nr contigs were blasted against the mosquito NCBI nr database to filter out Endogenous Viral Elements (EVEs; generally below 1000 bp in our dataset and not abundant but highly prevalent in almost all mosquito samples of the same *Aedes* species). Nr contigs with at least 70% nt similarity to mosquito sequences in the database were filtered out, resulting in 121 eukaryotic virus contigs. Abundance and taxonomy tables of these eukaryotic virus contigs were used for making comparative analysis: alpha diversity (Observed and Simpson indices), beta diversity (Bray–Curtis dissimilarity and ordination analysis with adonis2) and heat maps in R (v4.3.1) with ComplexHeatmap [38], ggplot2 [39], and phyloseq packages [40]. Also, an abundance correlation analysis was conducted in R to identify the un-annotated/mis-annotated segments of segmented viruses. This makes use of contigs corresponding to RNA-dependent RNA-polymerase (RdRp) to identify contigs that displayed significant correlations and similar prevalence across samples [21].

### 2.4. Virus Identification and Phylogenetic Analysis

For each identified eukaryotic virus contig, ORF Finder was utilized to predict Open Reading Frames (ORFs) and identify contigs with complete coding capacity. To determine the evolutionary history of potential novel viruses, phylogenetic trees were constructed based on amino acid sequences of the RdRp protein. Amino acid sequences from related viruses belonging to the same Family or Order were retrieved from NCBI and included in the analysis. Phylogenetic trees were constructed using the maximum likelihood method with 1000 bootstraps, and pairwise distances (%) were calculated in MEGA11 [41].

### 2.5. Aedes Mosquito Identification and Quantification of Selected Viruses

Following morphological identification, mosquito samples subjected to the qRT-PCR analyses were also molecularly identified via DNA barcoding. For each mosquito, individual homogenization was carried out in 400 μL of PBS, followed by centrifugation at 17,000*g* × for 3 min. From the resulting supernatant, 150 μL was utilized for DNA extraction using the QIAGEN Viral RNA mini kit without carrier RNA, following the manufacturer’s instructions, and eluted in a volume of 60 μL. The eluted volume (60 μL) was diluted to 120 μL and divided into 10 aliquots of 12 μL each and stored to avoid multiple freeze thawing cycles of large volumes. The first aliquot was used for the amplification of the cytochrome c oxidase subunit I region of *Aedes* mosquitoes, employing AUCOS primers (Appendix A) with the QIAGEN One-step RT-PCR kit (QIAGEN, Hilden, Germany), following the manufacturer’s protocol [42]. Briefly, the reaction mixture consisted of 10 μL RNase-free water, 5 μL of 5× QIAGEN OneStep RT-PCR Buffer, 1 μL of dNTP Mix, 1.5 μL of each Primer (10 μM), 1 μL of QIAGEN OneStep RT-PCR Enzyme Mix, and 5 μL of sample, in a total volume of 25 μL [41]. Thermal cycling conditions for 35 cycles were as follows: Initial PCR pre-denaturation step at 95 °C for 10 min, denaturation at 94 °C for 30 s, annealing at 52 °C for 40 s, extension at 72 °C for 45 s, and a final extension at 72 °C for 10 min. PCR products were evaluated for amplification using 1.5% agarose gel electrophoresis in 1× TBE buffer and Midori Green as the intercalating agent and thereafter visualized under UV light to observe a band of ~700 base pairs (bp). Afterward, the amplicons underwent Sanger sequencing at Macrogen (Amsterdam, The Netherlands).

From NGS results obtained from the 54 sequenced pools, a total of seven viruses were selected based on abundance across samples and completeness of their genome sequences. Primers and probes were designed for qPCR quantification targeting the most conserved regions using NCBI Primer-BLAST (Appendix A). The remaining aliquots of the individual mosquitoes (182 single mosquitoes) were analyzed using each of the seven developed qRT-PCRs. Each qRT-PCR reaction was carried out in a total volume of 20 μL, consisting of 5 μL TaqMan Fast Virus 1-Step Master Mix by Thermo-Fisher (Waltham, MA, USA), 2 μL forward and reverse primers (10 μM each), 1 μL probe of (5 μM), and 5 μL nucleic acid extract. Standards (representing the qRT-PCR target regions) of known concentration (integrated DNA Technologies) were serially diluted in ten folds (10^3^–10^8^ copies) to make a standard curve. Quantification of genome copies for each virus was later determined by multiplying the Cq value obtained from qRT-PCR by the dilution factor 64.08. This factor accounts for the initial dilution in PBS (400 μL), followed by extraction (150 μL taken), and elution in 120 μL of elution buffer. The dilution factor was calculated, considering the respective dilution factors for PBS (2.67) and elution buffer (24), resulting in an overall dilution factor of 64.08.

## 3. Results

### 3.1. Each Sampling Site is Dominated by a Single Aedes Species

This study focuses on analyzing the eukaryotic virome of the *Aedes* mosquito from four regions in southwestern Cameroon. In 2020, a total of 398 *Aedes* mosquitoes were captured from Bafoussam (*n* = 101), Edea (*n* = 96), Buea (*n* = 96) and Yaounde (*n* = 105) (Figure 1). Analyses of the mosquito distribution revealed that *Ae. albopictus* species predominated in Edea, Buea, and Yaounde, while *Ae. africanus* was the most prevalent in Bafoussam. Small numbers of *Ae. simpsoni* were also captured, co-existing in regions predominated by *Ae. albopictus*. Only a single *Ae. aegypti* mosquito was captured in Edea (Figure 1).

### 3.2. Distinct Virus Families Identified within Aedes Mosquito Species from Different Sampling Sites

Across all four locations, 54 pools of *Aedes* mosquitoes were sequenced which yielded 121 eukaryotic viral contigs. These eukaryotic viral contigs were identified and annotated, revealing their closest relatives to be 37 distinct eukaryotic viruses. Up to 11 of these 37 eukaryotic viral genomes were closely related to known virus species which belonged to a group without an official family classification, while the 26 other viral genome sequences belonged to 16 established virus families (Figure 2a and Appendix A). Aside from these eukaryotic virus reads belonging to these 16 virus families, a significant proportion, accounting for 26.2% of all eukaryotic virus reads were attributed to the viral genomes not classified within established families (Figure 2b and Appendix A).

### 3.3. The Majority of Eukaryotic Viral Genomes are Found in Aedes africanus Species Collected in Bafoussam

There was a notable disparity in the distribution of eukaryotic reads across *Ae. africanus* and *Ae. albopictus* pools. Approximately 80.8% of all eukaryotic virus reads were identified in *Ae. africanus* pools and only 19.2% in *Ae. albopictus* pools (Figure 3a). The 80.8% found in *Ae. africanus* were all from the Bafoussam sampling site. *Ae. albopictus* mosquitoes were captured in the three other regions, namely Yaoundé, Buea, and Edea, and constitutes 13.0%, 6.2% and 0% of all eukaryotic reads respectively (Figure 3b and Appendix A). A significant proportion of these eukaryotic virus reads belonged to viral genomes which have not yet been assigned to established virus families (26.1% in *Ae. africanus* pools and 0.07% in *Ae. albopictus* pools) (Appendix A).

### 3.4. Significant Difference in Eukaryotic Virome Richness and Diversity between Aedes Mosquito Species

The alpha diversity of the eukaryotic viromes of *Ae. africanus* and *Ae. albopictus* populations circulating in all four locations showed a significant discrepancy in the richness and diversity (Figure 4a). Specifically, the virome of *Ae. africanus* was more abundant and diverse compared to that of *Ae. albopictus*. For the regions where *Ae. albopictus* was captured and found to contain eukaryotic viruses (Buea and Yaoundé), there was no statistically significant difference in the richness and diversity of these viromes (Figure 4b). Intriguingly, despite Edea sharing the same climatic zone with Buea, eukaryotic virus contigs were not identified in *Ae. albopictus* mosquito pools from Edea.

### 3.5. Significant Difference in Eukaryotic Virome Composition and Distribution of Aedes Mosquito Species from Different Sampling Sites

Beta diversity based on Bray–Curtis dissimilarity showed a distinct partitioning of eukaryotic virus communities at the level of mosquito species rather than the sampling sites (Figure 5). This shows that the specificity of the mosquito species is the dominant driving force in shaping the virome composition across different habitats.

### 3.6. Diverse and Abundant Eukaryotic Virome in Aedes africanus Compared to Aedes albopictus Mosquito Pools

The results obtained from the alpha and beta diversity analyses are also reflected on the log2 normalized abundance heatmap of the eukaryotic virus species in 54 pools (Figure 6). The abundance of viral reads per species is shown by the intensity of the red color on the heatmap. Notably, eukaryotic virus contigs related to 37 known viruses were identified in this study, showing a very distinct virome for each *Aedes* species. The virome of *Ae africanus* was observed to be more abundant and diverse (30 unique virus species) compared to that of *Ae. albopictus* (6 unique virus species). Interestingly, one viral genome was shared between both the *Aedes* species, showing distant (41.7% BLASTx) resemblance with Hattula totivirus 3. For each virus species, the average BLASTx percent is represented by the different shades of blue on the left in Figure 6, indicating that most of the identified viruses only showed rather low amino acid similarities to viruses present in GenBank. Furthermore, the dendrogram in the heatmap highlights that the clustering of the viromes is mainly driven by mosquito species and, to a lesser extent, by location.

### 3.7. Phylogenetic Analysis of Six Novel Viruses Identified in Aedes Mosquito Pools

Near complete genome sequences were identified for six novel viruses (BLASTx < 90% with viruses in databases), and these were used for phylogenetic analyses (Appendix A). All of these viruses were found in *Ae. africanus* samples collected in Bafoussam in 2020. Figure 7 presents a graphical illustration of the RdRp gene of these novel virus genomes, their closest relatives, as well as their pairwise similarities. In addition, Figure 7 shows phylogenetic trees constructed based on the amino acid sequence of the RdRp, which is the most conserved region in these viral genomes. The names of the novel viruses are indicated in red while their closest relatives are in blue. Also, each tree depicts the clade to which novel viruses belong, color-coded in red.

#### 3.7.1. *Bunyavirales*

*Bunyavirales* is an order encompassing negative sense single-strand, enveloped RNA viruses. Our tree focuses on three families, namely *Phenuivirdae*, *Peribunyaviridae*, and *Nairoviridae*. Two novel viruses with three segments each (encoding for the S, M, and L proteins.) were identified and named Bafoussam mosquito bunyavirus 1 (BMBV1) and Bafoussam mosquito bunyavirus 2 (BMBV2). The L sequences of these two viruses (BMBV1 and BMBV2) had a 68% pairwise amino acid similarity to each other. BMBV1 and BMBV2 showed 44% and 45% pairwise similarities, respectively, with the L sequence of Salarivirus Mos8CM0 (Figure 7a). These similarities are reflected in the phylogenetic tree, where BMBV1 and BMBV2 are both located in a sub-clade, distant from the closest relative Salarivirus Mos8CM0. These 3 viruses cluster most closely to members of the family *Phenuiviridae*, which are known to infect mammals, birds, arthropods, plants, and fungi. A notable member is the arbovirus Rift valley fever virus.

#### 3.7.2. Orthomyxoviridae

*Orthomyxoviridae* is a family of negative sense single-strand, enveloped RNA viruses with eight segments. Two novel viruses were identified—Bafoussam mosquito orthomyxovirus 1 (BMOV1) with seven segments and Bafoussam mosquito orthomyxovirus 2 (BMOV2) with eight segments. For both viruses, segments encoding key viral proteins, including Polymerase basic protein 1 (PB1), Polymerase basic protein 2 (PB2), Polymerase acidic protein (PA), Nucleoprotein (NP), were identified. The PB2 sequences of BMOV1 and BMOV2 were highly distinct from each other, with only 53% pairwise amino acid similarity. However, the PB2 sequences BMOV1 and BMOV2 showed 55% and 78% pairwise similarity to that of Guadeloupe mosquito quaranja-like virus 1, respectively. Phylogenetically, both viruses (BMOV1 and BMOV2) fell in the same clade, although BMOV2 was more closely related to Guadeloupe mosquito quaranja-like virus 1, whereas BMOV1 formed an outgroup (Figure 7b).

#### 3.7.3. Rhabdoviridae

*Rhabdoviridae* is a family of negative sense single-strand, enveloped RNA viruses. The amino acid sequence of the L segment of the novel virus Bafoussam mosquito Rhabdovirus (BMRV) showed 59% pairwise amino acid similarity to Ohlsdorf ohlsrhavirus, which belongs to the genus *Ohlsrhavirus* within the family *Rhabdoviridae* (Figure 7c). Furthermore, the BMRV genomic sequence had five main ORFs, four of which coded for the proteins N, M, G and RdRp. However, ORF2 was very divergent and showed no amino acid similarity to the existing proteins in the database. After aligning this ORF2 amino acid sequence to the P-protein of Ohlsdorf ohlsrhavirus (occupying the same position of the P protein in the genome), we found a pairwise amino acid similarity of only 21%.

#### 3.7.4. Solemoviridae

*Solemoviridae* is a family of positive sense single-strand, non-enveloped RNA viruses. Bafoussam mosquito solemovirus (BMSV) had two segments identified with two ORFs each. The first segment had two ORFs coding for a hypothetical protein and the RdRp protein, while the second segment also had two ORFs coding for the Capsid protein and a hypothetical protein. The amino acid sequence of the first segment containing the RdRp protein of BMSV had a 23% pairwise amino acid similarity to that of the Pyongtaek culex solemovirus (Figure 7d).

### 3.8. In Search of an Aedes Mosquito Core Virome Using qRT-PCR

As NGS-based approaches provide only relative data rather than absolute quantification of viruses, we employed qRT-PCR to quantify the most abundant and prevalent viruses (identified in our mosquito pools) using individual mosquitoes. Additionally, qRT-PCR is known to be more sensitive compared to NGS in most instances. Quantification of these viruses was conducted in a total number of 182 individual *Aedes* mosquitoes (128 *Ae. albopictus*, 36 *Ae. africanus*, 17 *Ae. simpsoni*, and 1 *Ae. aegypti*) from Bafoussam (46 samples), Buea (41 samples), Edea (45 samples) and Yaoundé (50 samples), as summarized in Table 2. From *Aedes africanus*, we selected six abundant viruses as follows: BMSV, BMRV, BMOV1, BMOV2, BMBV1 and BMBV2, and the Guangzhou sobemo-like virus which was the most abundant in *Aedes albopictus* samples (Figure 8). Samples with Cycle threshold (Ct) values ≤ 35 indicating approximately 10 virus genome copies per mosquito were considered positive.

The Guangzhou sobemo-like virus was detected in samples from all four regions (Figure 8(ai,bi)). The samples from Buea had the highest median values for genome copies of 5.1 × 10^4^ in *Ae. albopictus* samples and 2.8 × 10^3^ in *Ae. simpsoni* samples. The Guangzhou sobemo-like virus was also prevalent in samples from Yaounde with median genome copies values of 3.7 × 10^4^ genome copies in *Ae. albopictus* samples and 1.3 × 10^3^ genome copies in *Ae. simpsoni* samples (Figure 8(ai)). In a total of 182 samples, 89 *Aedes* mosquito samples (48.9%) tested positive for the virus, and it was most prevalent in the *Ae. albopictus* samples (78/128), followed by the *Ae. simpsoni* samples (10/17). The virus was detected in only 1/36 *Ae. africanus* samples and absent in the only *Ae. aegypti* sample (Figure 8(bi)).

Bafoussam mosquito rhabdovirus (BMRV) was only detected in a few samples from only two regions, Edea and Yaoundé (Figure 8(aii)). Out of 182 *Aedes* mosquito samples, only 8 samples tested positive for the virus (4.4%). Compared to other *Aedes* species, BMRV was most prevalent in *Ae. albopictus* samples (7/128), followed by *Ae. simpsoni* samples (1/17) and absent in the *Ae. africanus* and *Ae. aegypti* samples (Figure 8(bii)).

Bafoussam mosquito solemovirus (BMSV) was present in samples from all four regions (Figure 8(aiii,biii)). Samples with the highest median viral loads were the *Ae. africanus* samples from Bafoussam (4.5 × 10^3^ genome copies), followed by the *Ae. simpsoni* samples from Edea (1.8 × 10^3^ DNA copies) (Figure 8(aiii)). BMSV was detected in 49/182 samples (26.9%), with the majority being in the *Ae. africanus* samples (25/36), followed by the *Ae. albopictus* samples (19/128), *Ae. simpsoni* samples (5/17) and absent in the *Ae. aegypti* samples (Figure 8(biii)).

Bafoussam mosquito bunyavirus 1 (BMBV1) was only identified in a few samples from Bafoussam (Figure 8(aiv,biv)). BMBV1 was detected in 14/182 samples (7.7%), with the majority being in the *Ae. africanus* samples (13/36). Only 1/128 *Ae. albopictus* samples tested positive for BMBV1, while the *Ae. simpsoni* and *Ae. aegypti* samples all tested negative for the virus (Figure 8(biv)).

Bafoussam mosquito bunyavirus 2 (BMBV2) was detected in samples from three regions, Bafoussam, Buea, and Edea (Figure 8(av,bv)). Only the *Ae. simpsoni* samples from Edea had a median genome copy number above zero (2.0 × 10^3^ genome copies) (Figure 8(av)). BMBV2 was identified in 17/182 samples (9.3%), most of which were in *Ae. albopictus* samples (13/128), followed by the *Ae. simpsoni* samples (3/17). The virus was only present in 1/36 *Ae. africanus* samples and absent in *Ae. aegypti* (Figure 8(bv)).

Bafoussam mosquito orthomyxovirus 1 (BMOV1) was only identified in samples from Bafoussam, just like the case of BMBV1 (Figure 8(avi,bvi)). This virus was present in 3/182 samples (1.6%). All three samples which tested positive for BMOV1 were the *Ae. africanus* samples (3/36) (Figure 8(bvi)).

Bafoussam mosquito orthomyxovirus 2 (BMOV2) was found in samples from three regions, Bafoussam, Buea and Edea (Figure 8(avii,bvii)). BMOV2 was identified in 16/182 samples (8.8%). Most of the samples which tested positive were the *Ae. africanus* samples (9/36), followed by the *Ae. albopictus* samples (7/128). The *Ae. simpsoni* and *Ae. aegypti* samples both tested negative for BMOV2 (Figure 8(bvii)).

## 4. Discussion

Mosquitoes are important vectors for pathogens like arboviruses which greatly influence human and animal health [21]. Our study focuses on the genus *Aedes* which constitutes one of the main vectors of arboviruses in Cameroon. *Ae. albopictus* and *Ae. aegypti* were described for a long time as the major vectors of arboviral diseases circulating in Cameroon, but recently, *Ae. africanus* mosquitoes were also found to be highly prevalent in West Cameroon [22,24,26]. In addition, *Ae. simpsoni* was recently identified to be abundant in rural forest settings in southwestern Cameroon and also in Maroua, located in North Cameroon [6,28].

These *Aedes* species are responsible for the circulation and transmission of arboviruses in Cameroon, causing arboviral diseases such as Dengue, Chikungunya, Yellow Fever, and Zika. Serological studies have reported the presence of these arboviruses in Cameroon by testing IgG and IgM antibodies (Dengue, Chikungunya and Zika) in the sera of inhabitants of Douala, Yaounde, Dschang, Garoua, Bertoua, Ngaoundere and Graoua [8,43].

In addition to being major vehicles of arboviruses in Cameroon, *Aedes* mosquitoes are also potentially home to ISVs which are not known to infect vertebrates, and hence do not cause diseases in humans and animals. In the last decade, research on ISVs has greatly increased due to their potential utility in the prevention and control of arboviral diseases. Mounting evidence shows that ISVs interact with other components of the mosquito microbiota and influence mosquito susceptibility to arboviral infection [27,43,44]. Unfortunately, very little is known about the virome of the *Aedes* mosquitoes circulating in Cameroon [26,41,42]

In this study, we employed viral metagenomics to characterize the viral composition of *Aedes* mosquito pools from four regions in the southwestern part of Cameroon representing three sub-climatic zones (Appendix A). In this part of the country, sampling showed that the dominant *Aedes* species was *Ae. albopictus* except for Bafoussam (neighboring town to Dschang), which was dominated by *Ae. africanus*. This invasive species has been reported to be more prevalent in the southern part of Cameroon because of the favorable climatic conditions which permit the proliferation of their eggs [22,27].

Among the 37 eukaryotic viruses identified in this study, 26 belong to established viral families. mNGS, despite its power, does not provide conclusive insights into the hosts of identified viruses in this study, limiting our ability to conclusively determine their origins. However, the identified viruses belonging to families known to infect mosquitoes and insects, such as the *Xinmoviridae*, *Iflaviridae*, and *Phasmaviridae*, likely represent true mosquito-infecting viruses [45,46]. Interestingly, we also identified viruses belonging to families containing known arboviruses, such as *Flaviviridae* (Menghai flavivirus) and *Peribunyaviridae* (Duke bunyavirus), suggesting a potential transmission risks to both arthropods and vertebrates, including humans. Moving on, we observed viruses from families possibly derived from the diet or the environment. Among these are *Anelloviridae*, *Circoviridae*, *Solemoviridae*, and *Totiviridae*, which may have been acquired from viremic hosts during blood feeding or from the environment during nectar feeding. Further, we identified eukaryotic viruses belonging to families known for infecting a broader range of hosts, including *Sedoreoviridae* (mammals, birds, arthropods, plants, algae), *Partitiviridae* (plants, fungi, protozoa), *Phenuiviridae* (mammals, birds, insects, plants, fungi), *Rhbadoviridae* (humans, animals, plants), and *Chrysoviridae* (fungi, plants, and possibly insects). Additionally, reads were detected that mapped to contigs annotated as Lampyris noctiluca errantivirus 1, a virus in the family *Metaviridae* known to infect animals, plants, and fungi. This family contains retrotransposons capable of inducing mutations and replicating via virus-like particles (VLPs). Moreover, eleven eukaryotic viruses not classified at the family level were identified, several of which were previously found in *Aedes* and *Ochlerotatus* species from various regions worldwide [47].

NGS data showed a striking difference in the eukaryotic virome of *Ae. africanus* and *Ae. albopictus* samples. The virome of *Ae. africanus* was richer and more diverse than the virome of *Ae. albopictus*. This could be due to environmental factors (breeding sites, sources of food), or host immune response to microbiota and microbiota interaction [48,49]. The rich and diverse microbiota of *Ae. africanus* could have a positive or negative effect on its susceptibility to arbovirus infection and transmission. The observation that the first isolation of ZIKV in mosquitoes was made in *Ae. africanus* [50], coupled with the fact that it is considered to be the main sylvatic vector of yellow fever virus in Africa [51], suggests that *Ae. africanus* is a competent vector for arboviruses. Unlike in other studies, where the virome of *Ae. albopictus* is dominated by more than one virus, in this study *Ae. albopictus* was dominated by one virus, Guangzhou sobemo-like virus [52,53,54,55]. The reason for this could be either methodological (differences in wetlab procedures or bio-informatics methods and used thresholds) or biological (difference in ISV carriage). In the case of the latter, this could have potential implications for distinct vector competences of Cameroonian *Ae. albopictus* mosquitoes versus mosquitoes in other regions.

Among the 37 eukaryotic viruses, we further characterized six novel viruses (BMSV, BMRV, BMOV1, BMOV2, BMBV1 and BMBV2) for which we obtained near complete genome sequences, all identified from *Ae. africanus* mosquitoes captured in Bafoussam [51]. Notably, these viruses showed a large genetic variation. Five of these novel viruses (BMSV, BMOV1, BMOV2, BMBV1 and BMBV2) clustered together in clades of unclassified viruses at the genus level indicating their unique evolutionary lineage and awaiting further official classifications. Only BMRV clustered within the established Genus *Ohlsrhavirus*. Although some of these newly identified viruses were found in families which contain Genera associated with human, animal, or plant diseases (*Orthomyxoviridae* and *Rhabdoviridae*), none of their closest relatives have demonstrated the ability to infect humans, animals, or plants, suggesting that they are all specific to insects.

To further investigate the concept of the “core virome” in mosquitoes [11,56,57], BMSV, BMRV, BMOV1, BMOV2, BMBV1, BMBV2, and the Guangzhou sobemo-like virus were quantified in 182 individual mosquitoes as they were abundantly present in our pools. The closest relatives of these seven (novel) viruses have all been previously detected in mosquitoes [11,52,54,55,57,58].

Among the seven viruses quantified in samples, only the Guangzhou sobemo-like virus and BMSV were found in samples from all locations (Figure 8; Appendix A). Although these viruses were found in samples from all locations, they were more abundant and prevalent in distinct *Aedes* species. The Guangzhou sobemo-like virus, first isolated from *Ae. albopictus* samples [57], was found in most of our *Ae. albopictus* and *Ae. simpsoni* samples, while BMSV was found in most *Ae. africanus* samples (Appendix A). In addition to BMSV, which was more prevalent in *Ae. africanus*, BMBV1 was mostly present in *Ae. africanus* samples. At the moment, a clear quantitative definition for a “core virome” is lacking but given the initial qualitative definition of “a set of viruses found in the majority of individuals in a particular mosquito population”, none of our identified viruses seems to meet this criterium for any of the investigated *Aedes* species across all the investigated sites in Cameroon. However, the Guangzhou sobemo-like virus and BMSV could be considered as core viruses in *Ae. albopictus*, and *Ae. africanus*, respectively, within particular restricted areas. In a recent study in Belgium, we were also unable to identify an abundant core virome in Culex mosquitoes [59]. For future research, it would be beneficial to have a more comprehensive and quantitative approach to compare mosquito virome compositions across species, space, and time.

Although our NGS data suggested a minimal overlap in the virome of *Ae. africanus* and *Ae. albopictus*, our more sensitive qRT-PCR data showed a larger overlap between the virome of both *Aedes* species (Figure 8). The qRT-PCR assays also showed a significant overlap in viruses found in *Ae. albopictus* and *Ae. simpsoni* (Figure 8).

For future research, it would be interesting to isolate the highly abundant and prevalent Guangzhou sobemo-like virus and BMSV for further studies and in vivo vector competence experiments. Understanding the complex interaction between these potential core viruses and the host and/or other components of the host microbiota is essential for gaining insights into the composition, diversity, and dynamics of the mosquito virome. This knowledge not only enhances our understanding of vector-borne disease dynamics but also helps in the development of more effective strategies for vector control and disease management.

## 5. Conclusions

Our study shows a striking difference (abundance and diversity) between the eukaryotic viromes of different *Aedes* species, with the virome of *Ae. africanus* being richer and more diverse from that of *Ae. albopictus*. We were unable to identify a true *Aedes* species specific core virome, although, on a local scale, Guangzhou sobemo-like virus and Bafoussam mosquito solemovirus virus could be considered as such in *Ae. albopictus* and *Ae. africanus*, respectively. Further studies are needed to understand if and how these viruses interact with the rest of the mosquito’s microbiota to influence vector competence.

## Figures and Tables

**Figure 1 viruses-16-01172-f001:**
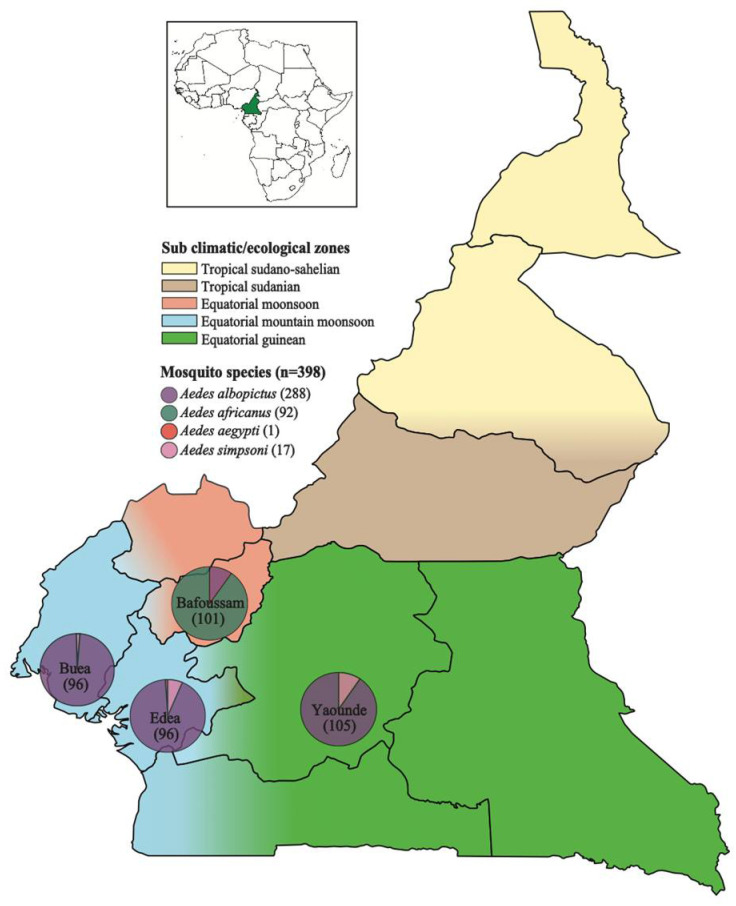
Map showing the climatic zones in Cameroon and the collection sites of *Aedes* mosquitoes in the southwestern part of Cameroon. The pie charts show the proportion of the different *Aedes* mosquito species found in each region.

**Figure 2 viruses-16-01172-f002:**
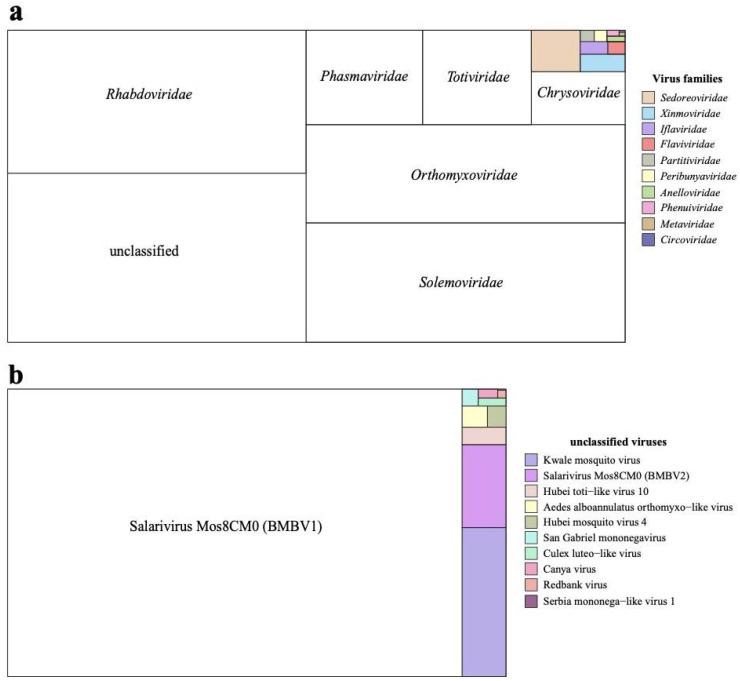
Proportion of eukaryotic virus reads assembled from 54 mosquito pools for each virus family. (**a**) Proportion of eukaryotic virus reads for each viral family. (**b**) Proportion of eukaryotic virus reads for viruses unclassified at family level.

**Figure 3 viruses-16-01172-f003:**
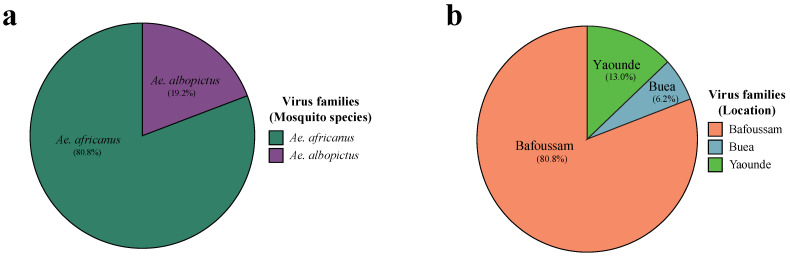
Proportion eukaryotic virome reads of *Aedes* mosquito species from the southwestern part of Cameroon. (**a**) Eukaryotic virome reads of *Ae. africanus* and *Ae. albopictus* (**b**) Eukaryotic virome reads of *Aedes* mosquitoes from Yaoundé, Buea, and Edea.

**Figure 4 viruses-16-01172-f004:**
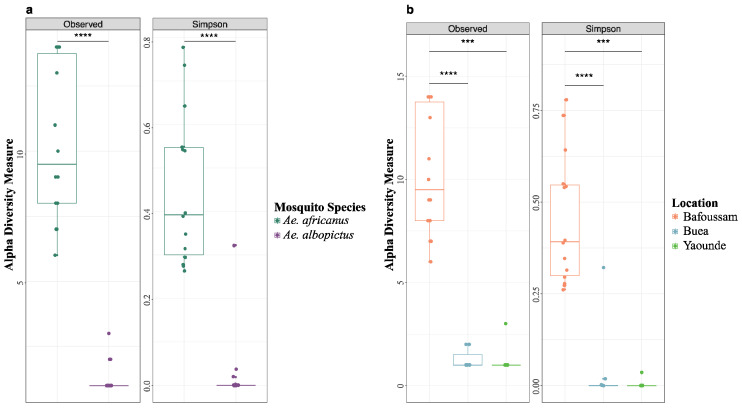
Alpha diversity of eukaryotic viruses from pools of *Aedes* mosquito. (**a**) Alpha diversity comparison between *Ae. africanus* and *Ae. albopictus*. (**b**) Alpha diversity comparison between Bafoussam, Buea, and Yaoundé. Wilcoxon test: *p* < 0.0001 (****), *p* < 0.001 (***).

**Figure 5 viruses-16-01172-f005:**
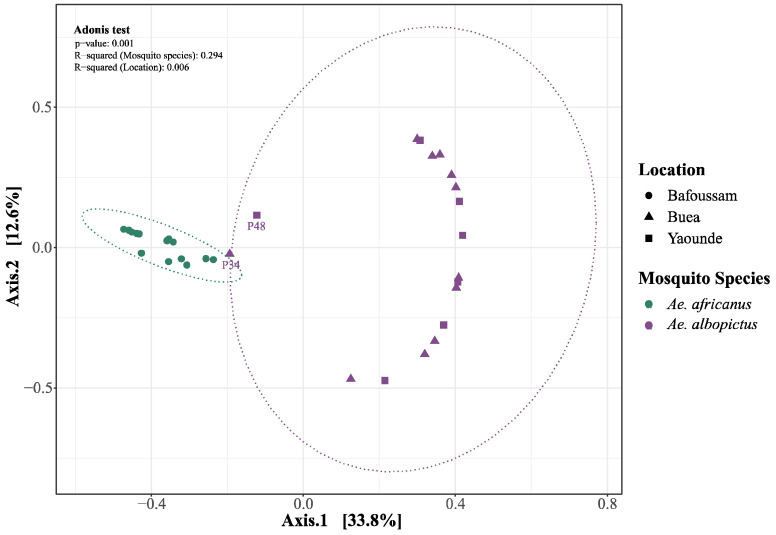
Principal Coordinates Analysis of eukaryotic viruses in *Aedes* mosquito species across all four locations (Bafoussam, Buea, Yaoundé). Adonis test: R^2^ (Mosquito species) = 0.294, R^2^ (Locations) = 0.006, *p* = 0.001.

**Figure 6 viruses-16-01172-f006:**
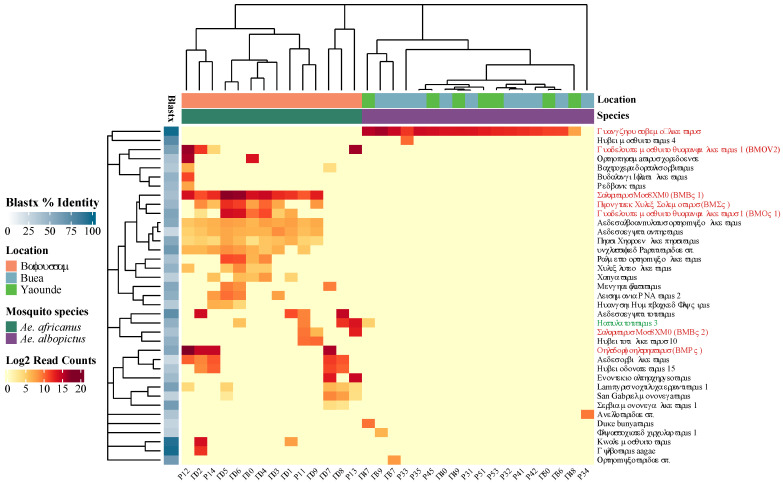
Read count of eukaryotic viral species on log2 scale. BLASTx percent identity to the most closely related reference sequence is shown in the shaded blue boxes. The virus name in green is the only virus species found in both *Aedes* species. Viruses in red were selected for qRT-PCR analysis and the abbreviations of novel viruses with near complete genomes (BLASTx < 90%) are shown within brackets.

**Figure 7 viruses-16-01172-f007:**
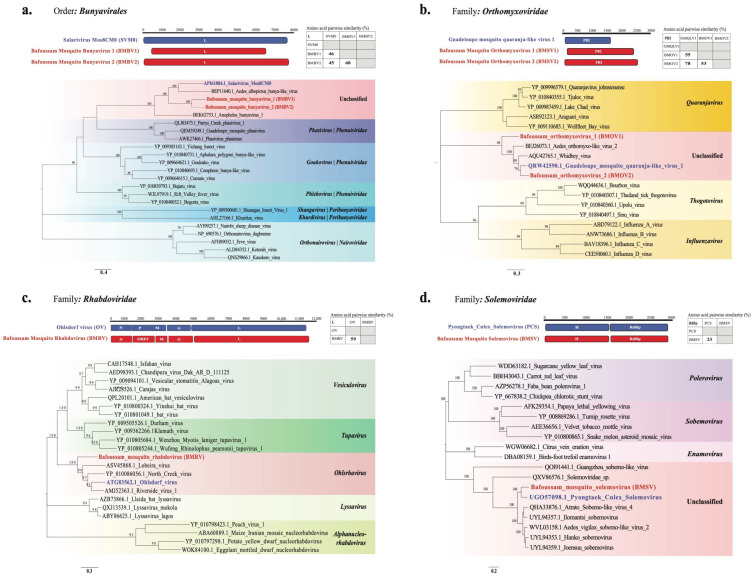
Genome organization, amino acid sequence similarity between novel virus and their closest relatives (novel virus in red and reference in blue), and maximum likelihood phylogeny based on amino acid sequence of PB2 or RdRp/L protein. (**a**) Bafoussam mosquito bunyavirus 1 (BMBV1) and Bafoussam mosquito bunyavirus 2 (BMBV2) (**b**) Bafoussam mosquito orthomyxovirus 1 (BMOV1) and Bafoussam mosquito orthomyxovirus 2 (BMOV2) (**c**) Bafoussam mosquito Rhabdovirus (BMRV) (**d**) Bafoussam mosquito solemovirus (BMSV).

**Figure 8 viruses-16-01172-f008:**
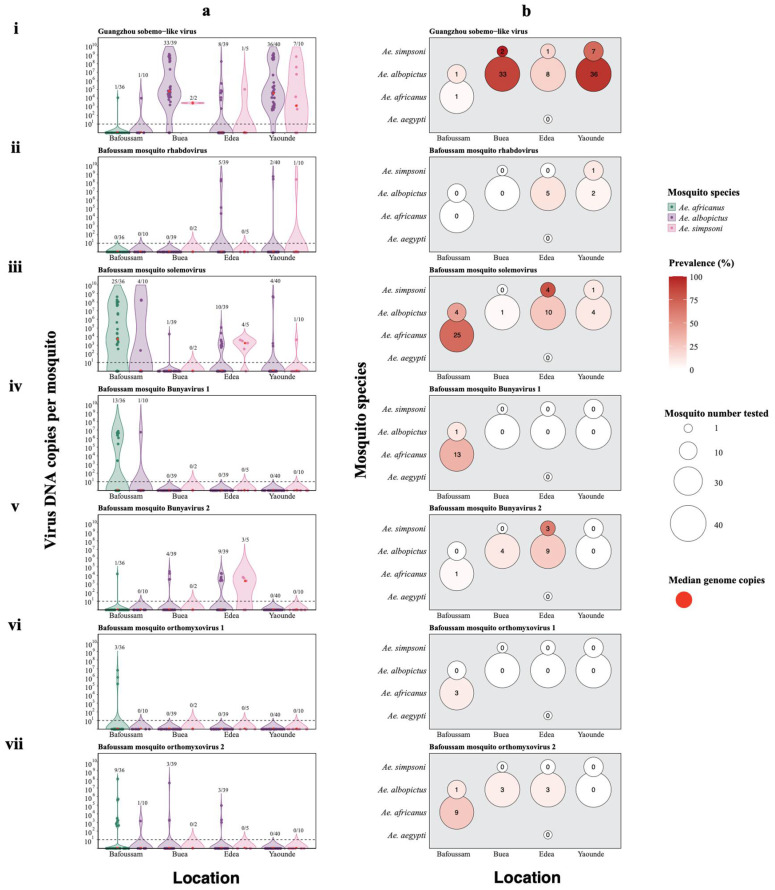
Quantification of seven abundant viruses. (**a**) Genome copy numbers of viruses in 182 individual mosquito samples. The red dot in the violin plot represents the median. The samples above the dotted line had Ct ≤ 35 and are thus considered positive. (**b**) Prevalence (%) of selected viruses across individual *Aedes* species is shown by intensity of red color. The values in the circles represent the number of mosquito samples that tested positive for the virus. (**i**–**vii**) Genome copies and prevalence of quantified viruses; Guangzhou sobemo-like virus, BMRV, BMSV, BMBV1, BMBV2, BMOV1 and BMOV2 respectively.

**Table 1 viruses-16-01172-t001:** Data on mosquito pools before Illumina sequencing.

Location	Mosquito Species	Number of Pools of 4 Mosquitoes
Bafoussam	*Aedes africanus*	14
Buea	*Aedes albopictus*	14
Edea	*Aedes albopictus*	12
Yaoundé	*Aedes albopictus*	14

**Table 2 viruses-16-01172-t002:** Data on individual mosquitoes for quantification of selected viruses.

Location	Mosquito Species	Individual Mosquitoes Tested
Bafoussam	*Ae. africanus*	36
	*Ae. albopictus*	10
Buea	*Ae. albopictus*	39
	*Ae. simpsoni*	2
Edea	*Ae. albopictus*	39
	*Ae. simpsoni*	5
	*Ae. aegypti*	1
Yaoundé	*Ae. albopictus*	40
	*Ae. simpsoni*	10

## Data Availability

The raw sequencing dataset for this study can be found through NCBI’s Sequence Read Archive (SRA) repository (Bioproject PRJNA1089369) and can be accessed as of 05 September 2024. Complete viral genome sequences were submitted to GenBank (PP764659-PP764664, PP868493-PP868501, PP898293-PP898302), and these sequences will be available as soon as processing is completed. All R scripts are available at https://github.com/Matthijnssenslab/CameroonMosquitoVirome.git.

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
