# Peer review of "Aedes Mosquito Virome in Southwestern Cameroon: Lack of Core Virome, But a Very Rich and Diverse Virome in Ae. africanus Compared to Other Aedes Species"

_viruses, 2024, doi:10.3390/v16071172_

Round 1

Reviewer 1 Report

Comments and Suggestions for Authors

The manuscript is well written in overall, and my comments are as below:

I currently do not have access to the scripts, GitHub is not working, please provide the link. 

Comments on the Quality of English Language

The manuscript is well written in overall, and my comments are as below:

Line 444: Remove extra space after comma in citation.

line 464:  "Sedoreoviridae" to "Seadornaviridae

line 531: "does not only enhance" to "not only enhances" 

Reviewer 2 Report

Comments and Suggestions for Authors

I have thoroughly reviewed your manuscript "Aedes mosquito virome in southwestern Cameroon: Lack of core virome, but a very rich and diverse virome in Ae. africanus compared to other Aedes species", and I must say I am truly impressed by its high quality overall.

The authors led a viral metagenomics study to characterize the viral composition of Aedes mosquito pools from four regions in the southwestern part of Cameroon, representing three sub-climatic zones. In total, they analyzed 54 pools of mosquitoes collected from 4 sampling sites in Cameroon. They found a significant difference in eukaryotic virome richness and diversity between Aedes mosquito species rather than sampling sites. Aedes africanus had 80.8% of all eukaryotic virus reads, while only 19.2% were detected in Ae. albopictus pools. 11/37 eukaryotic viral contigs were closed to known virus species, while 26/37 belonged to 16 other virus families. The also identified viral genomes not classified within established families. The manuscript is well written and has an interesting discussion covering the main points.  I have some questions and style aspects as follows:

Line 48: There are two periods.

Lines 485-487: The authors describe that “The reason for this could be either methodological (differences in wetlab procedures or bio-informatics methods and used thresholds), or biological (difference in ISV carriage).” Here, why do you hypothesize a potential effect of the methodological approach on the results? Do you run appropriate controls in your analysis to discard this potential bias? Did you include some controls in the lab analysis?

Could some viral sequences represent Non-retroviral Integrated RNA Virus Sequences (NIRVS)? Or how do you discard this potential explanation?

Round 2

Reviewer 1 Report

Comments and Suggestions for Authors

Could you please check the link and provide an updated URL or any additional instructions for accessing the scripts?

Author Response

Comments 1:

Could you please check the link and provide an updated URL or any additional instructions for accessing the scripts?

Response :
Thank you for your feedback. The link below allows full access to the R scripts used for the analysis of both NGS and qPCR data from the Mosquito Cameroon 2020 (MOSCAM20) project.

https://github.com/Matthijnssenslab/CameroonMosquitoVirome.git